# A Novel α_IIb_β_3_ Antagonist from Snake Venom Prevents Thrombosis without Causing Bleeding

**DOI:** 10.3390/toxins12010011

**Published:** 2019-12-21

**Authors:** Yu-Ju Kuo, Ching-Hu Chung, Tzu-Yu Pan, Woei-Jer Chuang, Tur-Fu Huang

**Affiliations:** 1Department of Medicine, Mackay Medical College, New Taipei City 25245, Taiwan; d01443001@ntu.edu.tw (Y.-J.K.); chchung@mmc.edu.tw (C.-H.C.); 2Graduate Institute of Pharmacology, College of Medicine, National Taiwan University, Taipei 10051, Taiwan; r02443003@ntu.edu.tw; 3Department of Biochemistry, National Cheng Kung University Medical College, Tainan 70101, Taiwan

**Keywords:** arterial thrombosis, antiplatelet agent, integrin α_IIb_β_3_, bleeding side effect, snake venom proteins, disintegrins

## Abstract

Life-threatening thrombocytopenia and bleeding, common side effects of clinically available α_IIb_β_3_ antagonists, are associated with the induction of ligand-induced integrin conformational changes and exposure of ligand-induced binding sites (LIBSs). To address this issue, we examined intrinsic mechanisms and structure–activity relationships of purified disintegrins, from *Protobothrops flavoviridis* venom (i.e., *Trimeresurus flavoviridis*), TFV-1 and TFV-3 with distinctly different pro-hemorrhagic tendencies. TFV-1 with a different α_IIb_β_3_ binding epitope from that of TFV-3 and chimeric 7E3 Fab, i.e., Abciximab, decelerates α_IIb_β_3_ ligation without causing a conformational change in α_IIb_β_3_, as determined with the LIBS antibody, AP5, and the mimetic, drug-dependent antibody (DDAb), AP2, an inhibitory monoclonal antibody raised against α_IIb_β_3_. Consistent with their different binding epitopes, a combination of TFV-1 and AP2 did not induce FcγRIIa-mediated activation of the ITAM–Syk–PLCγ2 pathway and platelet aggregation, in contrast to the clinical antithrombotics, abciximab, eptifibatide, and disintegrin TFV-3. Furthermore, TFV-1 selectively inhibits Gα_13_-mediated platelet aggregation without affecting talin-driven clot firmness, which is responsible for physiological hemostatic processes. At equally efficacious antithrombotic dosages, TFV-1 caused neither severe thrombocytopenia nor bleeding in FcγRIIa-transgenic mice. Likewise, it did not induce hypocoagulation in human whole blood in the rotational thromboelastometry (ROTEM) assay used in perioperative situations. In contrast, TFV-3 and eptifibatide exhibited all of these hemostatic effects. Thus, the α_IIb_β_3_ antagonist, TFV-1, efficaciously prevents arterial thrombosis without adversely affecting hemostasis.

## 1. Introduction

Integrin α_IIb_β_3_ is a member of the integrin family of adhesion receptors and remains in an inactive state in normal circulation, preventing undesirable thrombus formation [1]. Upon endothelial injury and agonist-stimulated platelet activation, the α_IIb_β_3_ complex undergoes a dramatic conformational change and assumes an active intermediate affinity state, which is recognized by the sequence, HHLGGAKQAGV, at the C terminus of the fibrinogen γ chain and Arg-Gly-Asp (RGD) sequences in the α chains of ligands [2].

Disintegrins are Arg-Gly-Asp (RGD)/Lys-Gly-Asp (KGD)-containing, cysteine-rich proteins in many snake venoms [3]. Based on the RGD/KGD mimetic sequence, disintegrins acts as α_IIb_β_3_ antagonists and potential antithrombotic agents [4,5,6]. Currently, three clinically available α_IIb_β_3_ antagonists, represented by abciximab, eptifibatide, and tirofiban, are used as potent antithrombotics for their rapid action and high efficacy. However, their use is primarily limited to patients undergoing percutaneous coronary intervention because of significant bleeding risk [7,8].

Life-threatening thrombocytopenia and bleeding, common severe side effects of these RDG-mimetic drugs and α_IIb_β_3_ antagonists [9,10], are associated with induction of drug-induced integrin conformational changes and ligand-induced binding site (LIBSs) exposure of integrin α_IIb_β_3_ [11,12]. Various reports indicate that thrombocytopenia after exposure to α_IIb_β_3_ antagonists is linked to drug-dependent antibody binding (DDAb) to platelets [10]. These antibodies are exquisitely drug-dependent and detect conformational changes in α_IIb_β_3_ elicited by antagonist binding. Drug-induced platelet activation in the circulation is associated with symptoms of disseminated intravascular coagulation due to platelet-activating antibodies [13,14]. These antibodies are believed to form complexes with antagonists/platelet factor on the platelets, on endothelial cells, or in plasma, and the resulting ternary complex binds to platelet FcγRIIa (CD32) in an Fc-dependent fashion [15]. The complex induces FcγRIIa-mediated downstream activation of signaling components, resulting in platelet consumption and leading to thrombocytopenia and bleeding [16,17]. Therefore, at doses where current α_IIb_β_3_ antagonists exhibit efficacious antithrombotic activity, they also increase bleeding risk [18]. Thus, it is crucial to develop new antagonists that do not cause bleeding.

To address this issue, two disintegrins with different hemorrhagic properties, TFV-1 and TFV-3, were purified from *Protobothrops flavoviridis* venom. Previously, we found that eptifibatide or disintegrin TMV-2, in combination with 10E5 or AP2, the inhibitory monoclonal antibodies (mAbs) raised against α_IIb_β_3_, specifically triggered platelet activation [19]. Because of the unique properties of these mAbs, we used them as probes to clarify binding to α_IIb_β_3_ and the pro-hemorrhagic tendencies of disintegrins. TFV1, exhibiting a binding motif distinct from those of TFV-3 and abciximab, decelerated α_IIb_β_3_ ligation without causing a conformational change of integrin α_IIb_β_3_. At efficacious antithrombotic doses, TFV-1 prevents thrombus formation without increasing bleeding risk in the FcγRIIa transgenic mouse model, in contrast to TFV-3 and abciximab. Taken together, the pathological mechanism in α_IIb_β_3_ antagonist-induced thrombocytopenia and the structure–activity relationship of TFV-1 and TFV-3 may help to advance development of new, safer α_IIb_β_3_ antagonists with minimal effects on normal physiological hemostasis.

## 2. Results

### 2.1. Purification and Characterization of TFV1 and TFV3

Venom of *Protobothrops flavoviridis*, the Okinawa habu, was fractionated into three subfractions using an FPLC Superdex 75 column chromatography (Figure 1A). Fraction III (*) exhibiting potent platelet inhibitory activity was collected and further refractionated on a C_18_ reverse-phase HPLC column (Figure 1B). Two antiplatelet fractions eluted at approximately 10 and 20 min, and the dried purified proteins were lyophilized and named TFV-1 and TFV-3, respectively. Purified TFV-1 and TFV-3 behave as single-chain peptides by SDS-PAGE, since their mobilities remained the same in the presence or absence of 2% β-mercaptoethanol (Figure 1C). With MALDI-TOF, their molecular masses were 7310 and 7646 Da, respectively (Figure 1D,E).

To determine their sequences, high-energy collisional dissociation fragmentation was employed with liquid chromatography (LC)–tandem mass spectrometry (MS/MS). The results derived from top-down (Appendix A) and bottom-up approaches provided information on the sequences near the protein C- and N-termini, respectively. The partial sequence of TFV-1 exhibited 84% sequence identity with the flavostatin [20] (Figure 1F), a disintegrin purified from the venom of *Protobothrops flavoviridis*; and the partial sequence of TFV-3 had 80% sequence identity with trimestatin [21] (Figure 1F), another disintegrin from the same venom.

### 2.2. Aggregation Studies of the RGD-Containing Disintegrins, TFV-1 and TFV-3

In human platelet-rich plasma (PRP), both TFV-1 (Appendix A) and TFV-3 (Appendix A) caused concentration-dependent inhibition of platelet aggregation induced by collagen (10 μg/mL) or adenosine 5′-diphosphate (ADP, 20 μM). In human platelet suspensions (PS), TFV-1 (Appendix A) and TFV-3 (Appendix A) blocked the platelet aggregation caused by collagen (10 μg/mL) or thrombin (0.1 U/mL) in a concentration-dependent manner (Table 1). These results indicated that TFV-1 and TFV-3 inhibit platelet aggregation by blocking an essential step of platelet aggregation with a similar IC_50_ irrespective of agonists used.

### 2.3. TFV-1 Binds to an Epitope of Integrin α_IIb_β_3_, Different from Those Bound by TFV-3 and Abciximab

To examine the binding motif of TFV-1 and TFV-3 toward the α_IIb_β_3_ receptor, we used FITC-TFV-1 (Figure 2A) or FITC-TFV-3 (Figure 2B) as probes to examine the effects of a monoclonal antibody (mAb) raised against the α_IIb_β_3_ domain, on binding of disintegrins to platelets. It is well established that the binding site of mAb 7E3 is near the RGD binding site of the βA-domains [22], whereas mAb 10E5 interacts with the cap subdomain of the α_IIb_ β-propeller [12]. FITC-TFV-1 binding to α_IIb_β_3_ was competitively inhibited by mAb 10E5, but not mAb 7E3, while FITC-TFV-3 binding to α_IIb_β_3_ was competitively blocked by mAb 7E3 but not mAb 10E5 (Figure 2A,B).

We previously reported that mAb 7E3 shares the same binding site with RGD-containing α_IIb_β_3_ antagonists rhodostomin and trigramin [5,23], which cause thrombocytopenia and bleeding owing to their effects on a conformational change of integrin α_IIb_β_3_. Since the humanized version of a function-blocking mAb, c7E3 (i.e., abciximab) has been reported to bind to the βA domains and subsequently induces exposure of ligand-induced binding sites and consequent thrombocytopenia [9,24], we used abciximab as a positive control (Figure 2C). Interestingly, we found that TFV-3 competitively inhibited mAb 7E3 binding to platelet α_IIb_β_3_, while TFV-1 did not affect binding of mAb 7E3. Furthermore, TFV-1 competitively reduced binding of mAb 10E5 to platelets, while abciximab and TFV-3 did not (Figure 2D). Together, these data demonstrated that the RGD-bearing disintegrins TFV-1 and TFV-3 inhibit agonist-induced platelet aggregation via α_IIb_β_3_ receptor blockade. Furthermore, the binding site of TFV-3 is close to the βA domains and similar to that of abciximab, while the binding site of TFV-1 is near the α_IIb_β_3_-propeller domain.

### 2.4. TFV-1 Binding to Integrin α_IIb_β_3_ Does Not Prime the Resting α_IIb_β_3_ to Bind Ligand

Immune thrombocytopenia occurs on first exposure to RGD-mimetic agents. That is, platelet count usually declines sharply within hours of the commencement of drug administration, demonstrating the presence of a naturally occurring antiplatelet antibody in patients who took these kinds of drugs [11]. Previous reports have revealed that upon binding of RGD-mimetic drugs to integrin α_IIb_β_3_, the ligand-binding capacity increased in the activated integrin and intrinsic antibodies recognized conformational changes in α_IIb_β_3_ induced by drugs [12]. Thus, we tested the ‘priming’ effect of these α_IIb_β_3_ antagonists. In this assay, the ability of agents to induce resting integrin α_IIb_β_3_ to adopt a high-affinity ligand binding conformation was judged by measuring the intensity of fluorescence-conjugated fibrinogen (Figure 3A) or PAC-1 (Figure 3B) binding to platelets [25], and compared with platelets without agent administration. TFV-3, RGD-mimetic agent eptifibatide, and abciximab increased fibrinogen (Figure 3A) or PAC-1 (Figure 3B) binding to platelets. In contrast, TFV-1 did not prime integrin α_IIb_β_3_ to high-affinity state, demonstrating that TFV-1 does not induce major conformational changes in the integrin β_3_ subunit or prime the receptor.

### 2.5. TFV-1 Does Not Cause a Conformational Change of α_IIb_β_3_ Identified by LIBS Antibody AP5 or mAb AP2

Binding of RGD mimetic to integrin α_IIb_β_3_ causes conformational changes of α_IIb_β_3_ and exposure of cryptic epitopes, termed LIBS (ligand-induced binding sites), which are recognized by mAbs PMI-1, AP5, and D3 [26,27]. Therefore, we investigated the effect of TFV1 and TFV3 on the conformational change of α_IIb_β_3_ using the LIBS antibody, AP5, raised against conformation-dependent epitopes (Figure 3C,D) or mAb AP2 raised against α_IIb_β_3_ [28] (Figure 3E,F) as a platform. After treatment with TFV-3 and eptifibatide, the level of AP5 or AP2 binding to α_IIb_β_3_ was increased to 4–4.5× higher than in resting platelets, whereas TFV-1 did not enhance AP5 and AP2 binding to α_IIb_β_3_, compared with resting platelets.

### 2.6. Combination of TFV-1 with AP2 Does Not Induce FcγRIIa-Mediated Activation of the Downstream ITAM/Syk/PLCγ2 Pathway and Platelet Aggregation

We previously reported that upon drug-induced LIBS exposure, the mimetic drug-dependent antibody (DDAb) mAb, AP2, which was raised against the epitopes of α_IIb_β_3_, recruited FcγRIIa [19] and elicited FcγRIIa-mediated platelet aggregation [29] and platelet consumption [11]. The recent report also indicated that eptifibatide-induced thrombocytopenia is associated with an increase in circulating procoagulant platelet-derived microparticles [14]. Here, we found that rhodostomin, eptifibatide (1 μg/mL; Figure 4A), and TFV-3 (1 μg/mL; right panel of Figure 4B) showed a similar activating effect as when combined with AP2, while the combination of TFV-1 (1.5 μg/mL; left panel of Figure 4B) with AP2 did not induce platelet aggregation. Consistent with this finding, TFV-3 (2 μg/mL)/AP2 induced time-dependent phosphorylation of signal molecules, including focal adhesion kinase (FAK), Src, Syk, PI3K, and PLCγ2, while TFV-1 (2 μg/mL)/AP2 did not (Figure 4D). Pretreatment of FcγRIIa mAb (i.e., CD32), an ITAM-bearing transmembrane receptor responsible for α_IIb_β_3_ outside-in signaling [16], completely inhibited platelet aggregation caused by TFV-3/AP2 (Figure 4C), suggesting that FcγRIIa is essential for activation.

### 2.7. Combination of TFV-1 with AP2 Does Not Increase Intracellular Ca2+ Mobilization and P-selectin Expression in Human Platelets

FcγRIIa-mediated phosphorylation of ITAM tyrosine residues recruits and activates the tyrosine kinase, Syk, leading to intracellular Ca^2+^ mobilization, activation of MAPK/NF-κB pathways, and cell activation [30]. We found that pretreatment of Fura-2-loaded platelets with TFV-3 (1.5 μg/mL)/AP2 (4 μg/mL) increased cytosolic Ca2+ mobilization, while TFV-1 (1.5 μg/mL)/AP2 (4 μg/mL) did not (Figure 5A,B).

Upon platelet activation, P-selectin (CD62), a member of the C-type lectin family, is rapidly translocated from α-granules of platelets to the cell surface [31,32]. Therefore, P-selectin expression on the platelet surface has been widely used to characterize platelet activation [33]. Neither AP2 nor disintegrin alone could cause P-selectin expression (Figure 5C). However, the presence of TFV-3/AP2 (1/4 μg/mL) markedly increased expression of P-selectin, in contrast to TFV-1/AP2.

Shrinking and consolidation of fibrin clots are responsible for hemostatic processes and wound healing from within [34,35,36]; thus, we assessed the effect of TFV-1 and TFV-3 on thrombin-induced clot retraction in human PRP (Figure 6A,B). Like abciximab, TVF-3 also inhibited thrombin-induced clot retraction while TFV-1 did not, suggesting that TFV-1 has minimal effect on clot firmness during the hemostatic process.

To further mimic bleeding risk during perioperative scenarios, rotational thromboelastometry (ROTEM) [37,38] was used to evaluate platelet function in human whole blood by measuring the viscoelastic properties and kinetic changes of coagulation among these tested α_IIb_β_3_ antagonists (Figure 6C–H). In the INTEM assay (intrinsically activated test using ellagic acid), four ROTEM variables were taken as a representation of hemostasis, including the clotting time (CT), the clot formation time (CFT), the α angle, and the maximum clot firmness (MCF). These were incorporated into a coagulation index (CI) to provide an overall assessment of coagulation and clot firmness for physiological hemostasis. CT is an indicator of the initial coagulation rate. CFT and α angle are correlated to the kinetics and rate of clot formation. MCF represents maximal platelet–fibrin interaction. In human whole blood, TFV-3 and abciximab dose-dependently repressed kinetics of clot formation (Figure 6C), progressively delayed the clotting time, and reduced clot firmness (Figure 6D–G), as well as reducing the CI (Figure 6H), leading to hypocoagulation. At similar concentrations, TFV-1 showed fewer repressive effects compared with controls and showing that TFV-1 has less influence on primary hemostasis.

### 2.8. TFV-1 Exhibits Anti-Thrombotic Activity, but Reduced Tendencies to Cause FcγRIIa-Mediated Immune Clearance and Hemorrhage In Vivo

We assessed the in vitro antiplatelet activity of TFV-1 and TFV-3 in mouse PRP. Pretreatment of TFV-1 and TFV-3 inhibited collagen-induced platelet aggregation of PRP in concentration-dependent fashion with IC50s of 1.24 and 0.08 μg/mL (169 and 11 nM), respectively (Figure 7A). Next we administered tested agents to mice intravenously for 10 min and collected blood samples by intracardiac puncture (Figure 7B,C). Compared with the ex vivo inhibitory efficacy of abciximab and eptifibatide (Figure 7D), TFV-1 and TFV-3 revealed favorable inhibitory potency on platelet aggregation of PRP induced by collagen (Figure 7B–D) or ADP (Appendix A). Additionally, TFV-1 and TFV-3 did not affect the initial platelet shape change caused by these inducers. Furthermore, we investigated the in vivo antithrombotic activities of TFV1 and TFV3 in a ferric chloride (FeCl_3_)-induced arterial thrombosis model (Figure 7E,F). In response to 5% FeCl_3_-induced carotid arterial thrombosis, complete occlusion occurred in the control group within 9 min. Prophylactic intravascular injections of TFV-1 or TFV-3 prevented FeCl_3_-induced thrombus formation and delayed occlusion for over 80 min.

At an equally efficacious dose, we compared the in vivo effects of TFV-1 and TFV-3 with those of eptifibatide on platelet counts and bleeding times in a FcγRIIa-transgenic mouse model expressing FcγRIIa on platelets at equivalent levels to humans [39,40]. After intravenous injection of these tested agents into FcγRIIa transgenic mice, eptifibatide and TFV-3 not only caused descending platelet counts over 1–5 h (Figure 7G), but also prolonged tail bleeding time (Figure 7H) in a dose-dependent manner. In contrast, TFV-1 (0.25 or 0.5 mg/kg) did not affect platelet counts and bleeding times compared with the control group, suggesting that TFV-1 shows a lower tendency to cause FcγRIIa-mediated immune clearance of platelets and bleeding.

Since bleeding risks limit the use and doses of RGD-mimetics or α_I__I__b_β_3_ antagonists, we used AP2 to assess safety margins of these agents (Table 2). We administered AP2 with a 10× higher minimum effective dose of TFV-1 (15 μg/mL) and found that unlike low doses of TFV-3, abciximab, eptifibatide, or rhodostomin, TFV-1/AP2 does not cause platelet aggregation (Table 2). At 3-times higher minimum effective antiplatelet doses of these agents (Table 2), TFV-1 neither prolonged the clot formation time in the ROTEM assay nor showed a defect in thrombin-induced clot retraction in human PRP, in contrast to TFV-3, abciximab, and eptifibatide. These findings are consistent with the results of bleeding side effects in vivo. Consistent with these observations, TFV-1 has a wider safety margin/index (Table 2) than TFV-3 and these clinical antithrombotic drugs.

### 2.9. TFV-1 Does Not Affect the Interaction between Integrin α_IIb_β_3_ and Its Mediator Talin Responsible for the Hemostatic Process

Upon endothelial injury, platelet cytoplasmic talin drives inside-out signaling of α_IIb_β_3_, which is responsible for platelet adhesion to vessel walls and for clot formation in the initial hemostatic response [41]. Therefore, recent studies report that selectively targeting Gα_13_-mediated outside-in signaling without affecting talin-mediated integrin processes may provide a strategy for preventing thrombosis with a higher safety margin [42,43]. Thus, we further investigated effects of TFV-1 on the mutually exclusive binding of the integrin mediator talin and Gα_13_ to the β_3_-domain in human PS. TFV-1 selectively inhibited Gα_13_ binding to β_3_, but did not affect talin binding to the β_3_-domain, suggesting that TFV-1 acts as a selective inhibitor of Gα_13_ binding to β_3_ without perturbing talin-driven hemostatic processes (Figure 8).

## 3. Discussion

Clinically available α_IIb_β_3_ antagonists are highly efficacious antithrombotics, but their use is limited to percutaneous coronary intervention due to thrombocytopenia and bleeding [44]. Thrombocytopenia can occur on first exposure to RGD mimetic drugs and the platelet count declines abruptly within hours of commencement of drug administration [45], suggesting the presence of naturally occurring antiplatelet antibody [46]. Eptifibatide- or tirofiban-dependent antibodies (i.e., DDAbs) recognize ligand-induced binding sites (LIBS) in a drug-dependent manner [12,45,47], and subsequently recruit FcγRIIa and trigger a FcγRIIa-mediated platelet aggregation and consumption [19]. It has been reported that these intrinsic antibodies are capable of directly increasing in circulating procoagulant, suggesting that platelet activation might, in some instances, contribute to occasional thrombocytopenia [14]. However, the pathological mechanism by which such RGD mimetic drugs activate platelets remains unclear. Also, a potential antithrombotic agent with a higher safety profile is under active investigation.

We previously reported that the mAb, AP2, raised against conformation-dependent epitopes is used to mimic DDAb for predicting drug-induced platelet activation [29]. Here, we used AP2 as a platform to predict adverse reactions upon administration of RGD mimetic agents. Furthermore, we purified two small RGD-containing disintegrins, TFV-1 and TFV-3, both of which inhibited collagen-, ADP-, and thrombin-induced platelet aggregation in a concentration-dependent manner through α_IIb_β_3_ receptor blockade. Especially, we found that TFV-1 inhibited ligand binding to α_IIb_β_3_ by a mechanism different from that of clinically used α_IIb_β_3_ antagonists and TFV-3. Our results revealed that TFV-1, with a binding motif near the α_IIb_ β-propeller domain, which is different from that of TFV-3 binding near the βA domain, does not prime resting platelets to bind fibrinogen and PAC-1 (Figure 3A,B). While eptifibatide, abciximab, and TFV-3 significantly prime platelets to bind ligands, suggesting that TFV-1 exhibits the minimal intrinsic property of causing the conformational change of α_IIb_β_3_.

Upon conformational change, LIBS exposure occurs and DDAbs directly bind to conformation-altered α_IIb_β_3_ [10,12]. Consistent with these findings, the treatment of resting platelets with eptifibatide or TFV-3 caused a significant increase in LIBS antibody, AP5 (Figure 3C,D) and mAb AP2 (Figure 3E,F) binding, while TFV-1 did not, suggesting that TFV-1 causes minor induction of DDAbs with minimal effect on conformational change. In support of data also shown in Figure 4, the combination of TFV-1 with mAb AP2 did not induce FcγRIIa-dependent platelet aggregation or the time-dependent downstream ITAM/Syk/PLCγ2 pathway, which is associated with FcγRIIa-mediated platelet consumption and thrombocytopenia [11].

Since bleeding complications impose a major limitation on the clinical use of current anti-integrin and anti-thrombotic therapies [7], we evaluated bleeding contraindications of TFV-1 and TFV-3 at their efficacious antithrombotic dosage in vivo (Figure 7). Since mouse platelets do not express Fcγ receptors and lack the genetic equivalent of human FcγRIIa [39], in this study, an FcγRIIa transgenic mouse model [40] was used to better understand tendencies of these agents to trigger FcγRIIa-mediated thrombocytopenia. At efficacious antithrombotic doses, TFV-1 neither decreased platelet counts (Figure 7G) nor prolonged tail-bleeding time (Figure 7H) of FcγRIIa transgenic mice, whereas eptifibatide, abciximab, and TFV-3 had this adverse effect in vivo. Consistent with these results, TFV-3 and abciximab not only inhibited thrombin-induced clot retraction of human PRP (Figure 6A,B), but also affected all values associated with platelet function and blood coagulation in human whole blood (Figure 6C–H; ROTEM assay), resulting in hypocoagulation states. By contrast, TFV-1 had minor effects on these coagulation indexes. Overall, our data suggest that TFV-1 is a unique disintegrin, with minimal conformational effects, which could potentially prevent thrombosis without affecting physiological hemostasis.

## 4. Conclusions

In summary, the lead α_IIb_β_3_ antagonist, TFV1, with a specific binding motif, neither caused conformational changes in integrin α_IIb_β_3_ nor triggered FcγRIIa-mediated activation of the downstream ITAM/Syk/PLCγ2 pathway. Therefore, TFV-1 exhibited anti-thrombotic activity, but lower tendencies to cause FcγRIIa-mediated immune clearance of platelets and hemorrhage. Furthermore, subsequent structure–activity studies of TFV-1 and TFV-3, and their interactions with α_IIb_β_3_ at a molecular level may provide valuable information for development of a second generation of α_IIb_β_3_ antagonists.

## 5. Materials and Methods

### 5.1. Materials

mAb 7E3 and 10E5 were kindly supplied by Barry S. Coller (Rockefeller University, New York, NY, USA). The mAbs, AP2 and AP5, were kindly donated by Peter J. Newman (Blood Research Institute, Blood Center of Wisconsin, Milwaukee, WI, USA).

Lyophilized crude venom of *Protobothrops flavoviridis* (i.e., *Trimeresurus flavoviridis*) was purchased from Sigma Chemical Co. (St. Louis, MO, USA) and stored at 4 °C. Fractogel EMD TMAE was purchased from Merck (Darmstadt, Germany). Fast protein liquid chromatography (FPLC) Superdex 75HR 10/300 columns were from Amersham Pharmacia (Uppsala, Sweden). A high-performance liquid chromatography C_18_ column was from Waters (Milford, MA, USA). Acetonitrile and trifluoroacetic acid (TFA) were from Merck (Darmstadt, Germany). Molecular-mass standards for electrophoresis were from Bio-Rad (Hercules, CA, USA). Pierce™ Glu-C Protease, MS Grade (90054), and Sulfo-NHS-biotin reagents were from Pierce (Rockford, IL USA). Acrylamide, adenosine diphosphate (ADP), bovine serum albumin (BSA), collagen (bovine tendon type I), Coomassie blue R-250, human thrombin, prostaglandin E1 (PGE1), sodium dodecyl sulfate (SDS), and tris-(hydroxymethyl) aminoethane HCl (Tris-HCl) were obtained from Sigma Chemical Co. (St. Louis, MO, USA). Heparin was from Leo Pharmaceutical Product (Ballerup, Denmark). FITC-conjugated goat anti-mouse IgG was from Santa Cruz Biotechnology, Inc. (Santa Cruz, CA, USA). Alexa 488-conjugated fibrinogen was from Invitrogen (Thermo Fisher Scientific Inc., Waltham, MA, USA). FITC mouse anti-human PAC-1 was from BD Biosciences (Becton, Dickinson and Company, CA, USA).

### 5.2. Top-Down Analysis

Purified and dried TFV-1 and TFV-3 samples were dissolved in 8 M urea/50 mM ammonium bicarbonate buffer, and disulfide bonds were reduced in 10 mM dithiothreitol at 37 °C for 1 h. LC-MS/MS analyses were performed with a Q Exactive™ mass spectrometer coupled with an UltiMate™ 3000 RSLCnano system (Thermo Scientific) using an Acclaim PepMap RSLC C_18_ column (75 μm I.D. × 15 cm, 2 μm, 100 Å). The following gradient was used: 1 to 50% B for 19.5 min, 50 to 60% B for 3 min, 60 to 80% B for 2 min, and 80% B for 10 min (0.1% FA as mobile phase A and 95% acetonitrile/0.1% FA as mobile phase B). MS scan range was set to be *m*/*z* 400–2500 at a resolution of 140,000 (FWHM). The mass spectrum for each protein was deconvoluted with Protein Deconvolution 4.0 software (Thermo Scientific) to obtain molecular weights. Precursor ions, *m*/*z* 813.80, 915.52, 1046.03, 1220.20, 1464.03, and 1830.04 for TFV-1 and *m*/*z* 955.05, 1091.48, 1273.06, and 1527.47 for TFV-3, were selected for targeted-MS^2^ analyses. Higher-energy collisional dissociation (HCD) at normalized collision energy of 32% was used to fragment these ions to generate multiplexed MS/MS spectrum for each protein. The mass spectra were inspected and relevant peaks were assigned manually.

### 5.3. Endoproteinase Asp-N or Glu-C Digestion and LC-MS/MS Analysis

Samples of TFV-1 and TFV-3 were denatured with 8 M urea and reduced in 10 mM DTT at 37 °C for 1 h. Alkylation was conducted using 50 mM iodoacetamide for 30 min in the dark at room temperature. Reduced TFV-1 and TFV-3 were diluted with 50 mM ammonium bicarbonate (Sigma-Aldrich Chemical Co.) and then digested with endoproteinase Asp-N (Roche) or Glu-C (Pierce) at 37 °C overnight, respectively. The same LC-MS/MS setup as described above was used for analyzing the peptide mixture. The following gradient was applied: 1 to 30% B for 39.5 min, 50 to 60% B for 3 min, 60 to 80% B for 2 min and 80% B for 10 min (0.1% FA as mobile phase A and 95% acetonitrile/0.1% FA as mobile phase B). Full MS scans were performed with a range of *m*/*z* 300–2000, and the 10 most intense ions from MS scans were subjected to fragmentation for MS/MS spectra. Raw data were processed using Proteome Discoverer 1.4 (Thermo Scientific) and a database search was performed using Mascot 2.4.1 (Matrix Science Inc., Boston, MA, USA) against the SwissProt database. Carbamidomethyl (C) was selected as a fixed modification and deamidation and oxidation were chosen as variable modifications. Mass tolerance windows were set as ±10 ppm and ±0.02 Da for peptide and fragment ions, respectively. Up to two missed cleavages were allowed for Asp-N digestion. All identified MS/MS spectra were manually confirmed to ensure quality.

### 5.4. Preparation of Human Platelets and Aggregation Assay

Blood was collected from healthy volunteers, who had not taken any medication within 2 weeks. All human participants provided informed consent, and this study was approved by the ethic committees and Joint Institutional Review Board (17-S-032-2), Medical Research Ethics Foundation, Taiwan. Preparation of human platelet-rich plasma (PRP) and platelet suspensions (PS) and platelet aggregation assay were performed as previously described [29].

### 5.5. Western Blotting and Immunoprecipitation

Washed platelets in an aggregometer cuvette (37 °C, 900 rpm) were treated with the tested agent in the presence of mAb AP2 or agonist thrombin. After incubation, platelets were lysed with lysis buffer (20 mM Tris-HCl buffer, pH 7.5, 150 mM NaCl, 1 mM EDTA, 1 mM EGTA, 1% Triton X-100, 2.5 mM sodium pyrophosphate, 1 mg/mL leupeptin, 1 mM β-glycerolphosphate, 1 mM Na_3_VO_4_, 1 mM PMSF). Insoluble materials were removed by centrifugation at 14,000 rpm for 15 min at 4 °C. Aliquots of cell lysates were resolved on 10% SDS-PAGE under reducing conditions and electrotransferred to Immobilon-PVDF membrane (Millipore). Western blotting and immunoprecipitation were conducted as described previously [19].

### 5.6. Binding Study

TFV-1 and TFV-3 were conjugated with FITC [48]. In brief, washed human platelets (3 × 10^8^ platelets/mL) containing 2 μM PGE_1_ were labeled with primary anti-α_IIb_β_3_ integrin mAbs 7E3 or 10E5 at room temperature (RT) for 30 min. Labeled cells were washed and then incubated with secondary FITC-TFV-1 or FITC-TFV-3 at RT for 30 min with a continuous shaking. After incubation, cells were washed, resuspended in PBS, and analyzed immediately by FACS Calibur. In addition, washed human platelets were incubated with TFV-1, TFV-3, or abciximab at RT for 30 min. Following incubation, platelets were washed and then labeled with mAbs 7E3, 10E5 or AP2 at RT for 30 min. Labeled cells were incubated with secondary FITC-conjugated goat anti-mouse IgG at RT for 30 min and then analyzed by FACS Calibur (Becton Dickinson, Franklin Lakes, NJ, USA).

### 5.7. Priming Assay

The priming assay was performed as described previously, with minor modifications [25,49]. Washed platelets in HEPES-modified Tyrode’s buffer were treated with eptifibatide, TFV-1, or TFV-3 for 30 min at room temperature. After incubation, washed human platelets were incubated with Alexa 488-conjugated fibrinogen or FITC-labelled PAC1 antibody for 30 min at 37 °C and analyzed by flow cytometry.

### 5.8. Definition of Safety Index

We defined a safety index = the lowest concentration of disintegrin to activate platelets in the presence of mAb AP2 (4 μg/mL)/ IC_50_ (μg/mL) of disintegrin for collagen-induced platelet aggregation in platelet suspension.

With eptifibatide, for example, in combination with 4 μg/mL AP2, the lowest concentration of eptifibatide to activate platelets was 1 μg/mL (Figure 4A). The IC_50_ of eptifibatide was about 0.52 μg/mL in collagen-induced platelet aggregation; therefore, we defined the safety index of eptifibatide as 1.92× to represent its safety margin in drug-induced platelet activation. The high safety index means that higher doses of disintegrin could cause platelet activation.

### 5.9. Clot Retraction

Clot retraction was conducted according to the method of Tucker [34] with minor modification. Briefly, agents were incubated with 200 μL PRP, 5 μL RBC (used to color the clot) at 37 °C for 1 min in glass tubes. After adding thrombin (4 U/mL), a sealed glass pipette was immediately placed in each tube and the kinetics of clot retraction were observed. Images were recorded at time 0 and every 15 min until 120 min. The ratio of clot retraction (%) was calculated by the volume of serum (test)/volume of serum (control).

### 5.10. Rotational Thromboelastometry (ROTEM)

Human whole blood was collected from healthy donors and the ROTEM assay was performed using the ROTEM^®^ delta system. Briefly, whole blood was treated with agents at 37 °C. Blood mixtures were transferred to a ROTEM plastic cup, and samples were recalcified with star-TEM reagent (0.2 M CaCl_2_) to initiate the INTEM assay (intrinsically activated test using ellagic acid). The speed at which a sample coagulates depends on the activity of plasma coagulation system, platelet function, fibrinolysis, and other factors. The clotting time (CT, sec), clot formation time (CFT, sec), alpha angle (α, o), and maximum clot firmness (MCF, mm) variables were taken as a representation of hemostasis, and could be incorporated into a coagulation index (CI) as defined by the equation: CI = −0.6516CT − 0.3772CFT + 0.1224MCF + 0.0759α − 7.7922. The CI functions as an overall assessment of coagulation, with values less than −3.0 said to represent a hypo-coagulable state and values over +3.0 said to be hyper-coagulable.

### 5.11. Animal Preparation

FcγRIIa-transgenic mice obtained from The Jackson Laboratory [40] (weighing 24–30 g) and the male ICR mice (weighing 20–30 g) were used in all studies. Animals were given continuous access to food and water under controlled temperature (20 ± 1 °C) and humidity (55% ± 5%). Animal experimental protocols were approved by the Laboratory Animal Use Committee of Mackay Medical College (A1060020).

### 5.12. FeCl_3_-Induced Arterial Thrombosis Model

Mice were anesthetized with sodium pentobarbital (50 mg/kg) by intraperitoneal injection, and then an incision was made with a scalpel directly over the right common carotid artery, and a 2-mm section of the carotid artery was exposed. A miniature Doppler flow probe was placed around the artery to monitor blood flow. Thrombus formation was induced by applying filter paper (diameter, 2 mm) saturated with 7.5% FeCl_3_ solutions on the adventitia of the artery. After 3 min exposure, the filter paper was removed and carotid blood flow was continuously monitored for 80 min after FeCl_3_ removal.

### 5.13. Tail-Bleeding Time

Mice were intravenously injected with agents via a lateral caudal vein. After injection for 5 min, a sharp cut 2 mm from the tip of the tail was made. The amputated tail was immediately placed in a tube filled with isotonic saline at 37 °C. Bleeding time was recorded for ≤10 min and the endpoint was the arrest of bleeding [50].

### 5.14. Statistical Analysis

Results were expressed as mean ± SEM. Statistical analysis was performed by one-way analysis of variance (ANOVA) and the Newman–Keuls multiple comparison test. A *p*-value less than 0.05 (*p* < 0.05) was considered as a significant difference.

## Figures and Tables

**Figure 1 toxins-12-00011-f001:**
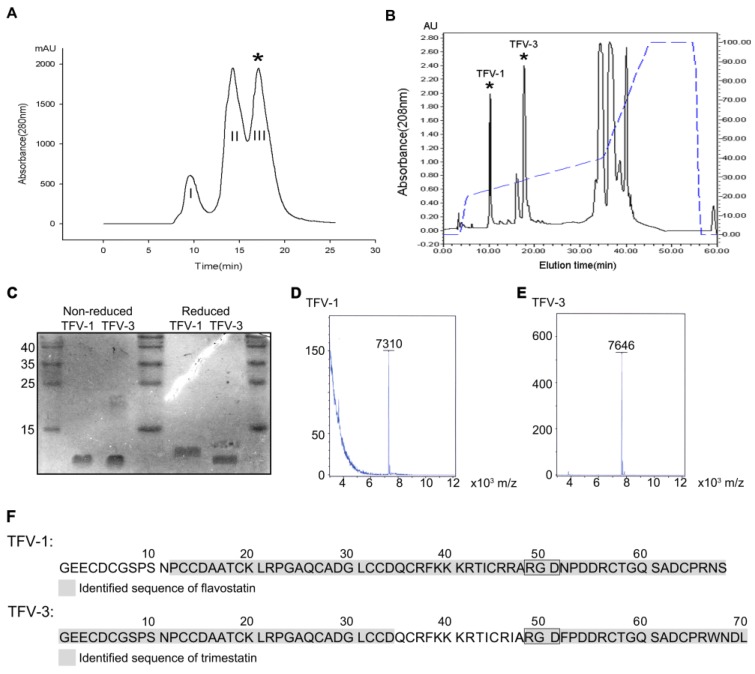
Purification and characterization of TFV1 and TFV3 from *Protobothrops flavoviridis* venom. (**A**) Purification of TFV1 and TFV3. 500 mg of crude venom was applied to a Superdex G-75 column. 0.01 N Ammonium bicarbonate in 0.15 N NaCl was used as the eluent at a flow rate of 0.75 mL/min. Fraction III (*, elution time ~15–17 min) exhibited potent inhibitory activity on collagen (10 μg/mL) and induced platelet aggregation. Therefore, this fraction was collected and further purified by reverse-phase HPLC. (**B**) Purification of TFV-1 and TFV-3 using reverse-phase HPLC. The antiplatelet fraction III (*) from the Superdex 75 column was applied to a C_18_ reverse-phase HPLC column equilibrated in 0.1% TFA at a flow rate of 0.8 mL/min. Chromatography was carried out with a two-solvent gradient (buffer A, 0.1% TFA in distilled water; buffer B, 80% acetonitrile with 0.1% TFA). Fractions were eluted over 60 min with a gradient of 0–80% acetonitrile (dashed line). TFV-1 eluted in approximately 24% acetonitrile at about 10 min. TFV-3 eluted in approximately 28% acetonitrile and an elution time of ~20 min. (**C**) TFV-1 and TFV-3 were run on 15% SDS-PAGE in the presence and absence of 2% β-mercaptoethanol. Gels were stained with Coomassie brilliant blue. Molecular masses of TFV-1 and TFV-3 were estimated at ~7 kDa. (**D**,**E**) MALDI-TOF mass spectra of TFV-1 and TFV-3 showed peaks with molecular masses of 7310 and 7646 Da, respectively. (**F**) Sequence determination of TFV-1 and TFV-3 using mass spectrometry. TFV-1 and TFV-3 sequences are marked in gray. Based on the MS/MS results, flavostatin was identified in sample TFV-1 (upper), while trimestatin was identified in sample TFV-3 (lower), which possesses a WNDL tetrapeptide at the C-terminus. The Arg-Gly-Asp (RGD) sequence common to both is indicated in a box.

**Figure 2 toxins-12-00011-f002:**
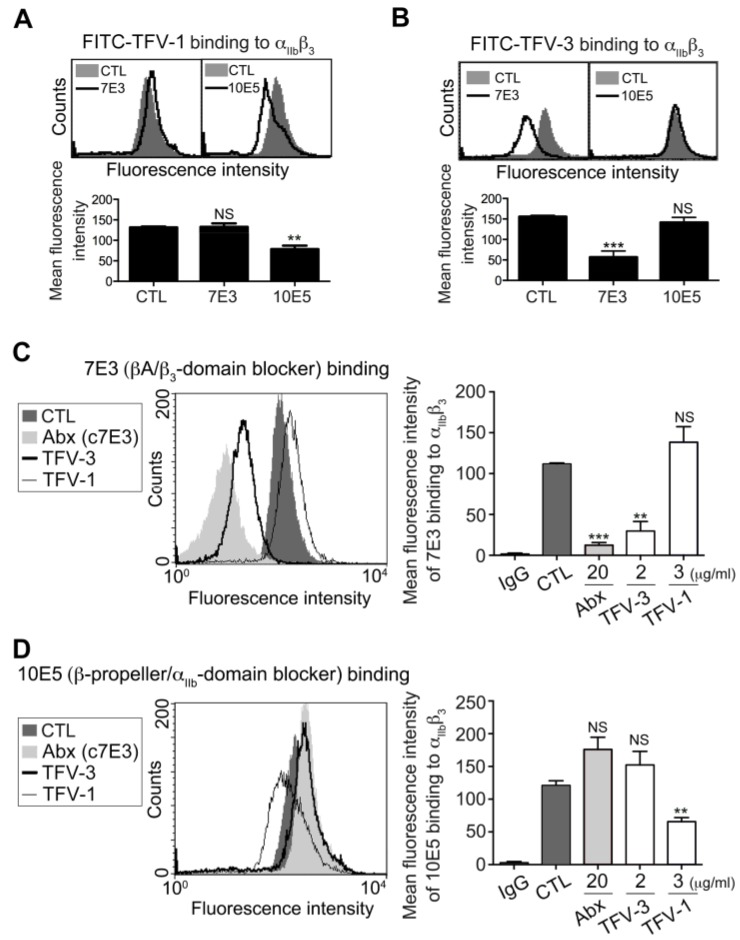
Effect of TFV-1 and TFV-3 on expression of integrin α_IIb_β_3_, probed by mAb 7E3 and 10E5 on unstimulated platelet. (**A**,**B**) Human PS was incubated with PBS (CTL), 7E3 ((**A**), 20 μg/mL), or 10E5 ((**B**), 20 μg/mL), probed with 5 μg/mL FITC-TFV-1 or FITC-TFV-3, as indicated, and subsequently analyzed by flow cytometry (mean ± SEM, n ≥ 10, * *p* < 0.05, ** *p* < 0.01, *** *p* < 0.001 compared with control group by Dunnett’s test; NS, non-significance). (**C**,**D**) Human PS was incubated with PBS (CTL), abciximab, TFV-3, or TFV-1, and then probed with 20 μg/mL mAb 7E3 (**C**) and 10E5 (**D**) raised against α_IIb_β_3_. Finally, the expression of mAb binding to α_IIb_β_3_ was analyzed by flow cytometry using FITC-conjugated anti-IgG mAb as a secondary antibody (mean ± SEM, error bars, n ≥ 8, ** *p* < 0.01, *** *p* < 0.001 compared with control group by Dunnett’s test; n.s, non-significance).

**Figure 3 toxins-12-00011-f003:**
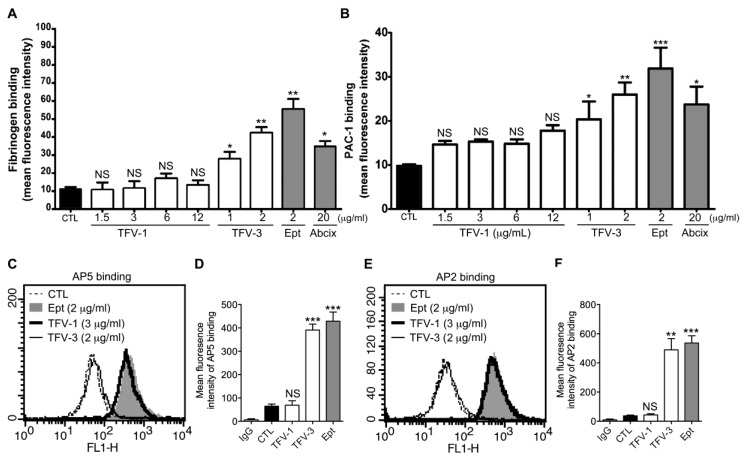
Effect of TFV-1 and TFV-3 on the conformational change of α_IIb_β_3_, ligand-induced binding sites (LIBS) exposure, and intrinsic monoclonal antibody recruitment. (**A**,**B**) Eptifibatide (Ept), abciximab, or various concentrations of TFV-1 or TFV-3 were added to human PS and then the platelets were fixed with 1% paraformaldehyde. After quenching the paraformaldehyde with glycine and washing, fluorescent fibrinogen ((**A**), 200 μg/mL) or PAC-1 ((**B**), 10 μg/mL) was added for 30 min at 37 °C. After washing twice, bound fluorescent fibrinogen or PAC-1 was detected by flow cytometry. The data shown are the mean fluorescence intensity of platelets in the presence of each compound. (mean ± SEM, error bars, n ≥ 5, **p* < 0.05, ***p* < 0.01 and ****p* < 0.001 compared with the control group by paired Newman–Keuls test; NS, non-significance) (**C**–**F**) Washed human platelets were incubated with PBS (Ctl), Eptifibatide (Ept, 2 μg/mL), TFV-1 (3 μg/mL) or TFV-3 (2 μg/mL), and then probed with 20 μg/mL mAb AP5 (**C**,**D**) or AP2 (**E**,**F**) raised against the ligand-induced binding site and the intrinsic antibody binding site of α_IIb_β_3_, respectively. Levels of AP5 and AP2 binding to α_IIb_β_3_ were analyzed by flow cytometry with fluorescein isothiocyanate-conjugated anti-IgG mAb as a secondary antibody. (mean ± SEM, error bars, n ≥ 6, ***p* < 0.01 and *** *p* < 0.001 compared with the control group by paired Newman–Keuls test; NS, non-significance).

**Figure 4 toxins-12-00011-f004:**
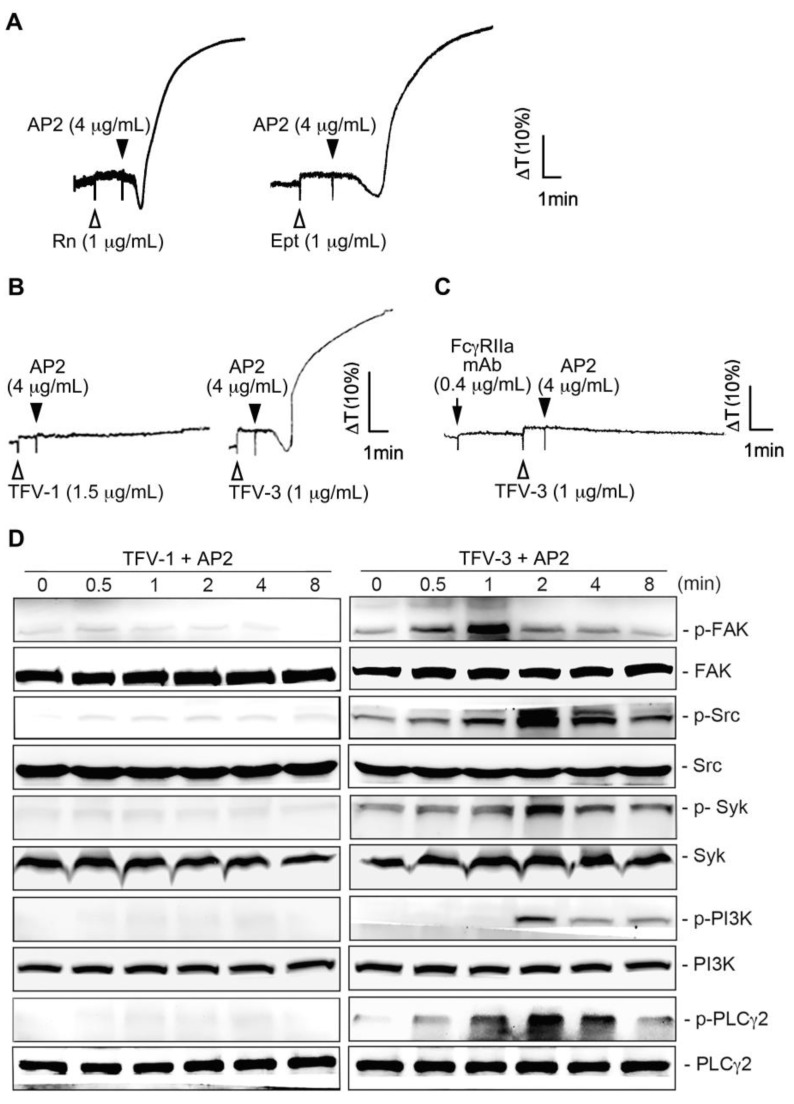
Effect of the combination of TFV-1 or TFV-3 with mAb AP2 on FcγRIIa-mediated downstream signaling and platelet aggregation. (**A**,**B**) Platelet suspension was incubated with rhodostomin ((**A**), left panel, Rn), eptifibatide ((**A**), right panel, Ept), TFV-1 ((**B**), left panel) or TFV-3 ((**B**), right panel) 1 min before addition of mAb AP2. (**C**) Human platelets were preincubated with anti-FcγRIIa mAb at 37 °C for 3 min, and TFV-3 and mAb AP2 were then added. These tracings in (**A**–**C**) are the representative ones that were reproducible at least three times. (**D**) Platelets were treated with TFV-1 (1.5 μg/mL)/AP2 (4 μg/mL) or TFV-3 (1 μg/mL)/AP2 (4 μg/mL) and aliquots were removed at indicated time periods. The extent of FAK, Src, Syk, PI3K, PLCγ2 phosphorylation was then determined by immunoblotting. Typical traces shown represent at least three independent experiments with similar results.

**Figure 5 toxins-12-00011-f005:**
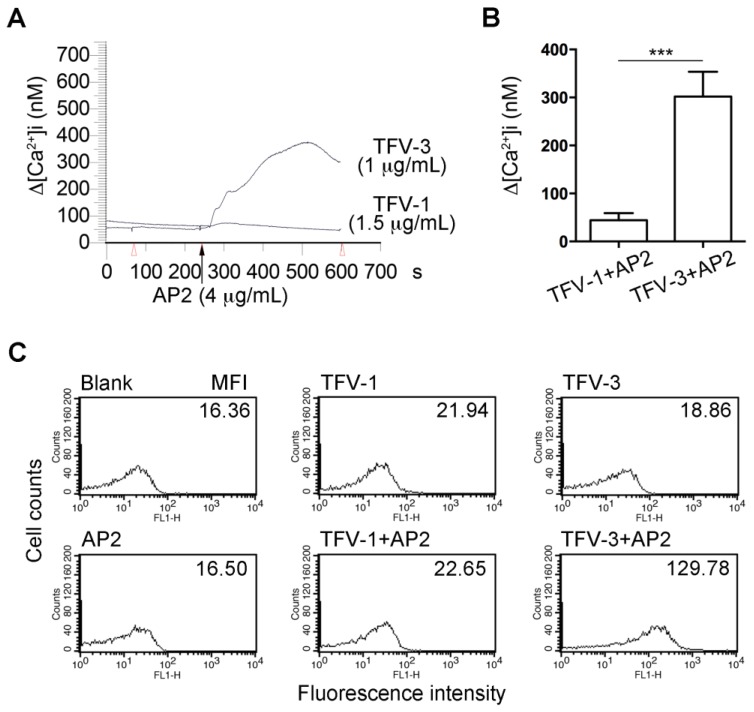
Effects of TFV-1/AP2 and TFV-3/AP2 on intracellular calcium mobilization and P-selectin expression in platelet suspension. (**A**,**B**) Fura-2 loaded platelets were resuspended in Tyrode’s buffer containing 1mM CaCl_2_, and the mAb, AP2 (4 µg/mL), was added to Fura-2-loaded platelets in the presence of TFV-1 (1.5 µg/mL) or TFV-3 (1 µg/mL), which were preincubated with platelets for 3 min. The level of [Ca^2+^]i was continuously monitored. (mean ± SEM, error bars, *n* = 5, *** *p* < 0.001 compared with the TFV-3/AP2 group by Dunnett’s test). (**C**) Flow cytometric analysis of P-selectin expression of platelet in the presence of TFV-1 (1 μg/mL), TFV-3 (1.5 μg/mL), AP2 (4 μg/mL), or the combination of TFV-1/AP2 or TFV-3/AP2. These experiments were repeated at least three times and only a representative tracing was shown.

**Figure 6 toxins-12-00011-f006:**
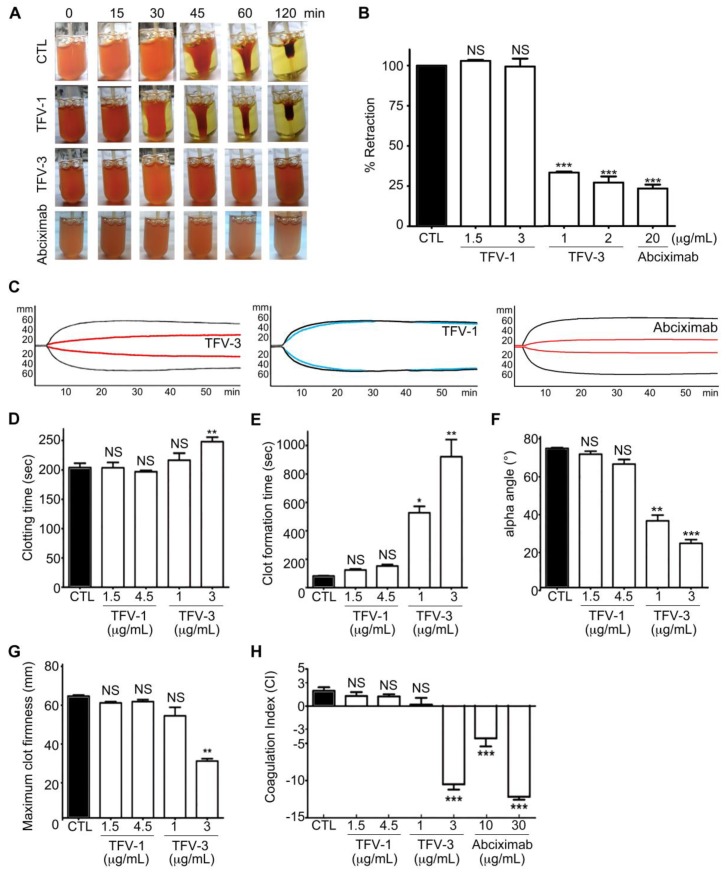
Effects of TFV-1 and TFV-3 on human hemostasis in vitro. (**A**,**B**) Effect of TFV-1 and TFV-3 on thrombin-induced clot retraction in human PRP. PRP was incubated with various concentrations of TFV-1 or TFV-3 at 37 °C for 3 min before addition of thrombin (4 U/mL). To observe the kinetics of clot retraction, photographs were taken at time 0 and every 15 min until 120 min. Percent retraction was measured by the volume of serum (test)/volume of serum (control). These data are presented as mean ± SEM (*n* = 6). *** *p* < 0.001 compared with the control group. (**C**–**H**) Physiologic platelet functions of TFV1 and TFV3 were evaluated by rotational thromboelastometry (ROTEM) assays. Human whole blood was incubated with TFV-1, TFV-3, or abciximab, and the ROTEM trace (**C**) of TFV-1 (1.5 μg/mL), TFV-3 (1 μg/mL) and abciximab (10 μg/mL) in human whole blood are shown. CTL (black line): in the absence of agents. Clotting time (**D**), clot formation time (**E**), α-angle (**F**), maximum clot firmness (**G**) and coagulation index (**H**) were evaluated with a ROTEM analyzer following recalcification of the blood, to determine clot and coagulation kinetics. Data were presented as means ± SEM (*n* = 7). * *p* < 0.05, ** *p* < 0.01, *** *p* <0.001 compared with the control group.

**Figure 7 toxins-12-00011-f007:**
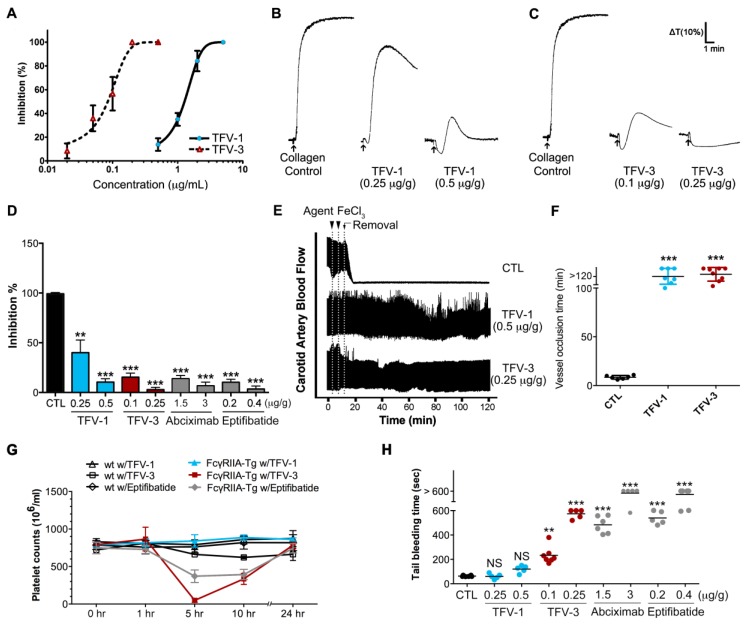
Ex vivo and in vivo effects of TFV-1: Selective inhibition of thrombosis, but not physiological hemostasis. (**A**) The in vitro collagen (10 μg/mL)-induced aggregation response of mouse PRP treated with TFV-1 or TFV-3. Concentration-dependent inhibition curves of TFV-1 and TFV-3 in mouse PRP are presented as mean ± SEM (*n* = 3). (**B**–**D**) Mice were intravenously treated with saline (control), TFV-1, TFV-3, abciximab, or eptifibatide for 10 min and then blood samples were collected by intracardiac puncture. PRP was obtained by centrifugation at 200× *g* for 4 min and then collagen (10 μg/mL) was added to trigger platelet aggregation. Platelet aggregation was measured by the turbidimetric method (ΔT) using a platelet aggregometer. Typical curves shown represent five independent experiments. Data are presented as a percent aggregation of control and mean ± SEM (*n* = 8). ** *p* < 0.01 and *** *p* < 0.001 compared with the control group (**E**,**F**) Antithrombotic activity of TFV-1 and TFV-3 in FeCl_3_-induced thrombus formation in mouse carotid artery. Mice were intravenously administered TFV1 or TFV3. After 5 min, FeCl_3_ injury was induced by a filter paper saturated with ferric chloride solution (10 %). After removal of the paper, carotid blood flow (mL/min) was monitored continuously until thromboembolism formation or for 60 min. Data are presented as the mean ± SEM (*n* ≥ 3). *** *p* < 0.001 as compared with the vehicle control (saline). (**G**) Effect of TFV-1 and TFV-3 on immune clearance of platelets in FcγRIIa-transgenic mice. Wild-type WT and FcγRIIa-transgenic mice were intravenously treated with TFV-1, TFV-3, eptifibatide, or abciximab, and then whole blood (100 μL) was collected by puncture of the retro-orbital sinus with heparinized hematocrit tubes. Platelet counts were obtained at timed intervals after injection of antithrombotic agents. Mean platelet counts (±SEM) over time are shown. (**H**) Effect of TFV-1 and TFV-3 on tail bleeding time of FcγRIIa-transgenic mice. Bleeding times were measured 5 min after the intravenous injection of saline, TFV-1 or TFV-3 at the doses indicated. Bleeding times longer than 10 min were expressed as >10 min. The average bleeding time is indicated as (—). Each type of symbol represents the bleeding time of an individual mouse.

**Figure 8 toxins-12-00011-f008:**
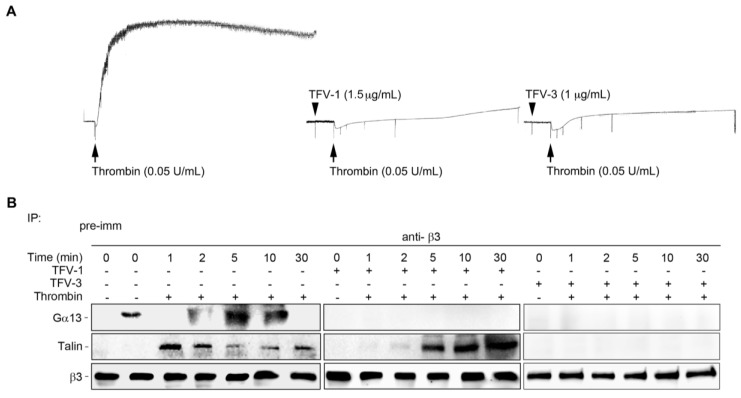
Comparison of the effect of TFV-1 and TFV-3 on the mutually exclusive binding of talin and Gα_13_ to cytoplasmic β_3_. Human platelets were stimulated with 0.1 U/mL α-thrombinin (Thr) in the presence of PBS, TFV-1 (2 μg/mL), or TFV-3 (1 μg/mL) with stirring (900 rpm) at 37 °C in an aggregometer and then solubilized at various time points. (**A**) Typical turbidity changes in human platelet suspension indicating integrin-dependent platelet aggregation. (**B**) Lysis of platelets by RIPA buffer that enables extraction and measurement of cytoplasmic proteins, including Gα_13_ and talin. Lysed platelets were immunoprecipitated with anti-β_3_ and immunoblotted for Gα_13_, talin, and β_3_. Tracings shown here were reproducible at least three times.

**Table 1 toxins-12-00011-t001:** IC_50_ of TFV-1 and TFV-3 in human PRP and human PS. PRP, platelet-rich plasma; PS, washed platelet suspension; ADP, adenosine 5′-diphosphate. The data are presented as means ± SEM (*n* = 5).

Antithrombotic Agents	TFV-1 (μg/mL)		TFV-3 (μg/mL)	
Inducer	PRP	PS	PRP	PS
ADP (20 μM)	0.720 ± 0.025		0.133 ± 0.003	
Collagen (10 μg/mL)	1.090 ± 0.148	0.530 ± 0.03	0.323 ± 0.124	0.217 ± 0.007
Thrombin (0.1 U/mL)		0.467 ± 0.02		0.301 ± 0.014

**Table 2 toxins-12-00011-t002:** IC_50_, safety indices, platelet hemostatic functions, and tail-bleeding time of TFV-1, TFV-3, and clinical anti-thrombotic agents.

Antithrombotic Agents	TFV-1		TFV-3		Abciximab		Eptifibatide		Control
The Dose of Agents (Fold of IC_50_)	2×	6×	2×	6×	2×	6×	2×	6×	-
IC_50_ (μg/mL)	0.74		0.45		5		0.52		-
Safety index	20.27		2.22		2.00		1.92		-
Clot formation time (s)	135 (NS)	172 (NS)	540 (*)	967 (**)	569 (***)	1355 (***)	518 (**)	622 (**)	105
Inhibition of Clot retraction (%)	0	0	77.3	86.1	54.7	92.5	63.2	89.4	-
Tail-bleeding time (s)	66.8 (NS)	102.5 (NS)	233.3 (**)	574.0 (***)	373.4 (***)	580.8 (***)	538.6 (***)	1341.2 (***)	68.7

IC_50_ of collagen (10 μg/mL)-induced platelet aggregation in washed platelet suspension. Safety index is estimated as the lowest concentration of disintegrin to activate platelet (combining with 4 μg/mL AP2)/IC_50_ of disintegrin on collagen-induced platelet aggregation. Inhibition of thrombin-induced clot retraction of human PRP (%) was measured by the volume of serum (agent−control)/volume of serum (control). Tail bleeding times of mice were measured at increasing antithrombotic dosages as compared with the control group (68.7 s, *n* = 28). (These experiments were repeated at least three times and values were presented as means. * *p* < 0.05, ** *p* < 0.01, *** *p* < 0.001 compared with control group by Dunnett’s test; NS, non-significance).

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
