# Peer review of "A Novel α_IIb_β_3_ Antagonist from Snake Venom Prevents Thrombosis without Causing Bleeding"

_toxins, 2019, doi:10.3390/toxins12010011_

Round 1
Reviewer 1 Report
Current clinically available αIIbβ3 antagonists are efficacious antithrombotics but have significant bleeding adverse effects. The main aim of the manuscript is to clarify the intrinsic mechanism in drug- induced thrombocytopenia and identify a potential candidate with minor bleeding risk.
Two disintegrins TFV-1 and TFV-3 have been purified from Trimeresurus flavoviridis snake venom and their structure-activity relationships have been characterized. TFV-1 exhibiting a specific binding motif, which is distinct from TFV-3, could decelerate αIIbβ3 ligation without causing a conformational change of this integrin.
Comparative study of TFV-1 and TFV-3 revealed:
TFV-1 and TFV-3 inhibit platelet aggregation through blocking a common step of platelet aggregation with a similar IC50 irrespective of different agonists used;
TFV-1 and TFV-3 inhibit agonist-induced platelet aggregation through αIIbβ3 receptor blockade;
TFV-1 competitively reduced mAb 10E5 binding to platelets while TFV-3 did not;
The binding site of TFV-1 is near to the αIIb β-propeller domain, while the binding site of TFV-3 is close to the βA domains;
TFV-1 and TFV-3 did not affect the initial platelet shape change caused by studied inducers in human platelets;
TFV-1 does not cause a conformational change of αIIbβ3 identified by LIBS antibody APS and mAb AP2;
TFV-1 inhibited ligand binding to αIIbβ3 by a mechanism different from that of clinical used αIIbβ3 antagonists and TFV-3.
The most important results are:
TFV-1 prevents arterial thrombosis without affecting physiological haemostasis of human platelet-rich plasma and whole blood in vitro;
TFV-1 neither caused severe thrombocytopenia and bleeding in FCγRIIa-transgenic mice nor induced a hypocoagulation state of human whole blood;
At efficacious antithrombotic dosage, TFV-1 potentially prevents thrombus formation without increasing bleeding risk in the FcγRIIa transgenic mouse model, in contrast to TFV-3 and abciximab.
Comments:
It is reasonable to add CONCLUSIONS
It is not clear how eleven amino acid residues of N-terminus of TFV-1 (see Fig.1) have been detected.
The parameters of theHPLC C18 column are not given.
There are mistakes in different sentences e.g lines 166-171, line 168-"who taken?", line 169- "drug binds".
The methods used in this study are adequate and appropriate for the resolving questions being addressed. The manuscript presents novel results to advance the field.
Reviewer 2 Report
The author report in this paper an interesting investigation on novel antagonists to alpha-IIb-beta-3 integrins, that may be useful in the prevention of thrombosis and bleeding. The work is complete and the results are reported in a clear and detialied manner.
I would suggest minor revisions as follows:
page 3 lines 113-116: the IC50 could be more easily read if reported in a table. Moreover a reference compound (abiciximab or eptifibatide as in the other tables) should be reported to validate the method. the author didn't report the method for FITC conjugation to their peptides page 7, the two paragraph are very short and could be merged page 9, notes to table 1: the method for safety index calculation should be here reported in detail, not only in the methods' section page 13, the last paragraph reporting the different effect of TFV-1 in signaling as selective inhibitor is the most important part of the paper, in my opinion because it represent a real novelty. This topic would deserve a larger space, may be moving the figure S2 into the main text. page 15: I suppose that the observational trial, involving human blood samples, needs to be registered in one of the public registries approved by the international health committes and the relative code should be reported in the manuscript. Moreover an ethic statement should be added.Author Response
Please see the attachment

Reviewer 3 Report
The authors report on a novel αIIbβ3 antagonist that has the potential to prevents thrombosis without causing hemorrhage due to thrombocytopenia. This information is of major interest. The study is carefully performed and the manuscript well written. However, there are several issues and concerns that require consideration by the authors to improve their manuscript.
Major points
The Discussion simply repeats the Introduction and Results sections and therefore requires complete revision. Specifically, the authors should address what the implication and significance of their findings are withe regard to future antithrombotic regimen. The authors previously reported that accutin/AP2 stimulation results in increased CD62P surface expression. Therefore, data on secretion from platelet α-granules upon TFV-1/AP2 or TFV-3/AP2 stimulation should also be presented. In platelets, ITAM/Syk/PLC signaling results in intracellular Ca2+ mobilization and subsequent extracellular Ca2+ influx. What about intracellular Ca2+ levels after TFV-1/AP2 or TFV-3/AP2 stimulation? Throughout, the numbers of experiments are low (n=3). It would strengthen the information, if the n-numbers are increased. The concentrations of antagonists indicated for the single experiments and corresponding figures are inconsistent, e.g., in Fig. 2/3 3 µg/ml TFV-1 and 2 g/ml TFV-3 are used, while in Fig. 5 the concentrations are half of that. Moreover, in Fig. 6 the agonist concentration is changed. Why? The authors should comment on this and provide an explanation for this discrepancy. 2 would be more informative when representative tracings of the aggregation experiments are displayed. Moreover, the authors should present data substantiating their statement (page 4, lines 123/124) that the platelet shape change is not affected by TFV-1 and TFV-3. In the legend to Fig. 1 (line 98/99), the authors indicate that fraction III inhibited GPVI-mediated platelet aggregation. However, no data at all are given. The authors present collagen- and ADP-induced aggregation of human platelets (Fig. 1). What about ADP-induced platelet aggregation of mouse platelets (Fig 7).Minor points:
In Fig. 5A, the shape change after eptifibatide treatment appears to be significantly prolonged compared to rhodostomin pretreatment. The authors should comment on this observation On page 2 (line 79), and in the legend to Fig. 1 (line 103), the authors state that TFV-1 was eluted after 15 min. Fig 1B clearly shows the indicated TFV-1 peak at approx. 10 minutes. Please correct or comment on this discrepancy. On page 5 (line 154), page 6 (line 190), and page 8 lines (233-235) signs/letters appear which are not defined/known. So please correct this. "Fig 5D" should be added to the sentence starting at line 216 and ending at line 218.Author Response
Please see the attachment
